# Molecular Characterization and Cross-Reactivity of Feline Calicivirus Circulating in Southwestern China

**DOI:** 10.3390/v13091812

**Published:** 2021-09-12

**Authors:** Long Zhou, Nengsheng Fu, Lu Ding, Yan Li, Jian Huang, Xue Sha, Qun Zhou, Xin Song, Bin Zhang

**Affiliations:** 1College of Animal & Veterinary Sciences, Southwest Minzu University, Chengdu 610041, China; zhoulongscu@163.com (L.Z.); fls17865690921@sina.com (N.F.); Ningning20002021@163.com (L.D.); ajhans@126.com (Y.L.); huangjian.1122@163.com (J.H.); ssx311@163.com (X.S.); qunyushan@yeah.net (Q.Z.); songxin116@sina.com (X.S.); 2Key Laboratory of Ministry of Education and Sichuan Province for Qinghai-Tibetan Plateau Animal Genetic Resource Reservation and Utilization, Southwest Minzu University, Chengdu 610041, China

**Keywords:** feline calicivirus, molecular characterization, phylogenetic analysis, cross-reactivity, neutralizing antibodies

## Abstract

Feline calicivirus (FCV) is an important pathogen of cats that has two genogroups (GI and GII). To investigate the prevalence and molecular characteristics of FCVs in southwestern China, 162 nasal swab samples were collected from cats in animal shelters and pet hospitals. In total, 38 of the clinical samples (23.46%) were identified as FCV positive using nested RT-PCR. Phylogenetic analyses using 10 capsid protein VP1 sequences revealed that 8 GI and 2 GII strains formed two independent clusters. Additionally, three separated FCVs that were not clustered phylogenetically (two GI and one GII strains) were successfully isolated from clinical samples and their full-length genomes were obtained. Phylogenetic and recombinant analyses of a GI FCV revealed genomic breakpoints in ORF1 and ORF2 regions with evidence for recombinant events between GI sub-genogroups, which is reported in China for the first time. Furthermore, sera obtained from mice immunized independently with the three FCV isolates and a commercial vaccine were used to evaluate the cross-reactivity of neutralizing antibodies. The three separate FCVs were neutralized by each other at a 1:19 to 1:775 titer range, whereas the triple-inactivated vaccine was at a titer of 1:16, which suggested that different genogroup/sub-genogroup FCV strains exhibit significantly different titers of neutralizing antibodies, including the commercial FCV vaccine. Thus, our study revealed the genetic diversity and complex cross-reactivity levels of FCVs in southwestern China, which provides new insights for application in vaccination strategies.

## 1. Introduction

Feline calicivirus (FCV), which belongs to the family *Caliciviridae*, genus *Vesivirus*, is the causative agent of respiratory disease, stomatitis, arthritis, and hemorrhagic-like fever in cats [1]. The virus was first isolated in 1957 [2] and is now widely distributed in many European, American, and Asian counties [3,4,5]. Generally, an FCV infection causes apparent or relatively mild oral cavity and upper respiratory tract symptoms [6,7]. Occasionally, FCV causes ulcerative dermatitis, severe pneumonia, feline chronic gingivo-stomatitis complex, abortion, acute jaundice, and acute febrile lameness syndrome [8,9]. However, in the past decade, virulent mutants of FCV called VS-FCVs, have been identified as the causal agents of severe and acute virulent systemic disease (VSD), resulting in a high persistent fever and high mortality rate in vaccinated cats [10,11].

The FCV genome is a non-enveloped, single-stranded positive-sense RNA of approximately 7.7 kb having three open reading frames (ORFs), including ORF1, ORF2, and ORF3 [12]. The ORF2 (2013 nt) encodes the capsid protein VP1 and is divided into six regions (A–F). The regions A, B, D, and F are relatively well conserved, whereas regions C and E are variable with significant sequence divergence levels [13]. Therefore, the *VP1* gene is frequently used as a target in molecular epidemiological investigations [4,14,15]. In accordance with the genetic diversity of the *VP1* gene, the global FCVs are classified into two genogroups: genogroup I (GI) and genogroup II (GII) [14]. In China, most FCV strains belong to GII, and they are closely related to Japanese FCV strains [5,16]. It is worthy to note that new FCV strains have been constantly emerging in recent years owing to the large degree of viral genetic variation [17,18,19].

Vaccination against FCV has been the mainstay control method for decades [20,21,22]. Although several commercial vaccines are widely used in the field and effectively reduce the clinical disease incidence, it is not clear whether the existing commercial vaccines have cross-protection against the newly emerging FCV strains. In the present study, 162 suspected FCV clinical samples were collected from nasal swabs from cats in southwest China from 2017 to 2020, and then the molecular characterization, virus isolation, and cross-reactivity of FCV strains were further investigated.

## 2. Materials and Methods

### 2.1. Sample Information and Treatment

A total of 162 nasal swab samples were collected from cats suspected to be infected with FCV owing to symptomology that included conjunctivitis, respiratory issues (such as coughing, sneezing), and nasal discharge. The cats were being treated in 2 pet hospitals and 14 animal shelters in the Chengdu and Chongqing Regions, southwestern China, from 2017 to 2020. The cats in the pet hospitals were life-long household animals. The shelters received cats from different places, and they were grouped in cages for a short period of time with veterinary care. The mean geographical distance among the collection sites was about 500 km including the two cities. The ages of the cats ranged from 2 months to 3 years old, and the weight ranged from 0.6 to 8.9 kg. All cats in this study were vaccinated with triple-inactivated feline Rhinotracheitis-Calici-Panleukopenia vaccines from Zoetis (Parsippany-Troy Hills, NJ, USA) or Boehringer Ingelheim (Ingelheim, Germany) (1 mL/dose) by intramuscular injection. According to the instructions of the vaccines, the two companies use the same strains, including feline rhinotracheitis virus (FRV-605 strain), feline calicivirus (FCV-255 strain), and feline panleukopenia virus (FPV-Cu-4 strain) as the triple-inactivated vaccine for production. Each sample was homogenized separately in phosphate-buffered saline (PBS) and centrifuged at 10,000 r/min for 10 min. The centrifugal supernatants were maintained at −80 °C until use.

### 2.2. Primer Design

One pair of nested RT-PCR primers was used to detect FCV from clinical samples as previously described [23], and another pair of primers (FCV-VP1-F/FCV-VP1-R) was designed for amplifying the *VP1* gene hypervariable (VP1 HV) region of FCV (Table 1). The complete genomic sequences of three FCV stains were amplified with six overlapping fragments (Appendix A) based on the full-length FCV reference sequences downloaded from NCBI GenBank.

### 2.3. RNA Extraction and RT-PCR for VP1 Gene

Viral RNA was extracted from sample homogenate with RNAiso Plus (TaKaRa, Dalian, China) according to the manufacturer’s instructions. The cDNA was reverse transcribed using a PrimeScript™ RT kit (TaKaRa, Dalian, China) following the manufacturer’s instructions. The VP1 HV gene of FCV strains in this study were amplified using the VP1 primer pair (Appendix A), and the PCR products were purified and cloned into pMD19-T vector (TaKaRa, Dalian, China). Then, the cloned viral cDNAs were sequenced by the Sanger sequencing approach (Sangon Biotech, Shanghai, China).

### 2.4. Virus Isolation and Complete Genome Amplification

The suspensions were passed through 0.22 μm filters and inoculated into the Crandell Reese Feline Kidney (CRFK) cell monolayer and cultured in DMEM supplemented with 5% FBS (HyClone, Logan, UT, USA) at 37 °C in a 5% CO_2_ incubator. Cell cultures were monitored every day for a cytopathic effect (CPE). If CPE appeared, the cells were freeze–thawed three times to release the virus. The viruses were further purified by plaque assay as previously described [16], and the purified viral isolates were cultured in monolayers of CRFK cells. The TCID_50_ of the FCV isolates were calculated using the method of Muench and Reed [24]. The supernatants of the CPE-positive cultures were negatively stained using 0.5% phosphotungstic acid (Thermo Fisher, Waltham, MA, USA) and observed by electron microscopy. The complete genomes of the FCV isolates were further amplified using the primers above (Appendix A). The sequences of six overlapping fragments from FCV isolates were assembled into full-length genome sequences using the SeqMan program in Lasergene software (DNASTAR, Madison, WI, USA).

### 2.5. FCV Sequences Analysis

Multiple sequence comparisons at nucleotide and amino acid levels were performed by the MegAlign program in Lasergene software7.2 (DNASTAR, Madison, WI, USA). Phylogenetic analysis of the VP1 HV gene and full-length FCV genomic sequences was performed using MEGA7.1 software (http://www.megasoftware.net, access time: 21 January 2021) (Tempe, AZ, USA) with the distance-based neighbor-joining method. The bootstrap values were calculated on 1000 replicates. All reference sequences used to construct phylogenetic trees in this study were downloaded from NCBI GenBank. To identify potential recombination events, the complete genomic sequences of FCV isolates were compared with the reference strains. The aligned nucleotide sequences of the complete genomes were analyzed using the SimPlot software (version 3.5.1) (JHK University, Baltimore, MD, USA) with a 500 bp window and a 20 bp step. In addition, phylogenetic trees based on each recombinant fragment were constructed to confirm the accuracy of recombination events.

### 2.6. Preparation of Antigen and Serum Virus Neutralization Assay for FCV Isolates

FCVs were propagated in CRFK cells and were inactivated with 1.0% β-propiolactone (Thermo Fisher, Waltham, MA, USA) for 8 h. Fifteen 6-week-old BALB/c female mice were randomly divided into five groups. Each group with 3 mice were immunized subcutaneously with condensed FCV-inactivated cell cultures mixed with Freund’s complete adjuvant and the commercial triple-inactivated vaccine (Zoetis), respectively, followed by two booster immunizations. The mice in the negative control group were inoculated with cell culture medium with Freund’s complete adjuvant. The serum virus neutralization assays were performed as follows: the sera were diluted 1:2, 1:4, 1:8, 1:16, 1:32, 1:64, 1:128, 1:256, 1:512, 1:1024, and 1:2048 by DMEM. The diluted sera were mixed with equal volumes of different FCV isolates (200TCID_50_) or commercial vaccine and incubated at 37 °C for 1 h. Then, the antibody–virus mixture was inoculated into CRFK monolayer cells, and the neutralizing antibody titers were obtained based on the estimation of 50% virus infection in cells as end points.

## 3. Results

### 3.1. RT-PCR Survey of Clinical Samples

In this study, 38 out of the 162 clinical samples were identified as positive for FCV using nested RT-PCR amplification, resulting in a positivity rate of 23.46% (38/162, 95% CI: 17.2–30.7%). Among them, 30 samples tested positive from 84 cats in 14 animal shelters (positive rate: 35.71%, 95% CI: 25.6–46.9%), and 8 samples tested positive from 78 cats in 2 pet hospitals (positive rate: 10.26%, 95% CI: 4.5–19.2%) (Table 1). Thus, the cats in the animal shelters had a significantly higher positive rate than those in the pet hospitals (*p* < 0.05).

### 3.2. Sequence Alignment and Phylogenetic Analysis of the VP1 Gene

Of the 38 positive samples, 10 VP1 hypervariable regions (HVs) were successfully obtained by sequencing and designated as SMU-B5, SMU-Y5, SMU-B9, SMU-B22, SMU-F4, SMU-Y14, SMU-C17, SMU-20, SMU-R1, and SMU-R55 (GenBank no.: MW194992–MW195001). Sequence alignment showed that the nucleotide similarity of the 10 VP1 HVs ranged from 72.5% to 92.8%, with 80.9–96.8% similarity levels in the deduced amino acid sequences. In addition, these viruses shared 72.0–89.2% nucleotide homology and 77.8–92.4% amino acid homology with the 35 reference FCV sequences from GenBank.

Phylogenetic analysis showed that the FCV strains clustered into two genogroups, GI and GII, as previously described [14], and the GI group is further classified into five clusters (I-V). Eight strains (SMU-B5, SMU-Y5, SMU-B9, SMU-Y14, SMU-C20, SMU-C17, SMU-R1, and SMU-F4) were divided into three clusters in the GI group. Among these strains, SMU-B5, SMU-Y5, and SMU-B9 are closely related to strain FCV-21223 from the USA, and SMU-Y14 and SMU-C20 share close genetic relationships with the strains HRB-SS and XH from China. Additionally, three strains (SMU-R1, SMU-F4, and SMU-C17) were classified into a minor branch together with the Chinese strains GX2019 and GX01–13 from Guangxi province. The remaining two strains (SMU-R55 and SMU-B22) belong to the GII group and are closely related to strain ym3/2001/JP from Japan (Figure 1). No strains in this study were divided into cluster II of the GI group, which contains the representative vaccine strains FCV-255 (KM111170), FCV-2024 (AF479590), and F9 (M86379), suggesting that the FCV strains identified in vaccinated cats certainly cannot be related to used vaccines. Thus, FCV strains present in southwestern China are genetically diverse and complex.

### 3.3. Isolation and Identification of FCV Strains

Four separated strains (SMU-B22, SMU-F4, SMU-Y14, and SMU-B5) that did not cluster phylogenetically were filtered out for further viral isolation, and three strains (designated SMU-B22-2020, SMU-F4-2020, and SMU-B5-2020) were successfully isolated from the clinical samples using CRFK cells. The infected CRFK cells exhibited cytogamy, cell rounding, karyopyknosis, and cluster-like forms; they eventually detached completely and remained suspended (Figure 2a, SMU-B22-2020 isolate shown only). No obvious CPE was observed in the control cells (Figure 2b). The three isolates were subjected to electron microscope observation after three rounds of plaque purification. Spherical viral particles with no envelopes were observed, and these features are consistent with the structural characteristics of caliciviruses (Figure 2c). The viruses were further conformed by RT-PCR using the FCV specific primers (Figure 2d). The TCID_50_ values of FCV SMU-B22-2020, SMU-F4-2020, and SMU-B5-2020 were determined as 1 × 10^9.5^/0.1, 1 × 10^11^/0.1, and 1 × 10^8.5^/0.1 mL, respectively.

### 3.4. Full-Length Genome Sequence Analyses of Three New FCV Isolates

To further understand the genetic variation of the FCV strains in southwestern China, the full-length or nearly full-length genomes of three novel strains were successfully amplified from the positive FCV samples. The results have been submitted to the GenBank database (accession no.: MW194989–MW194991), and showed 7653 bp, 7686 bp, and 7705 bp in length, respectively. The genetic analyses using MegAlign software revealed that the nucleotide identity ranged from 76.8% to 80.9%, and the sequences shared 76.1–82.9% nucleotide identity levels with 31 other full-length genome reference sequences.

To examine the genomic characteristics of the three FCV strains, the nucleotide (nt) and amino acid (aa) homology levels of the FCV ORFs among the three strains in this study and with the 31 representative strains were compared (Figure 3a,b). For SMU-B22-2020, ORF1 and ORF3 shared 83.7–86.6% nt (93.5–94.5% aa) homology levels with CH-JL1, which was the highest homology level among the reference strains, whereas ORF2 of SMU-B22-2020 shared a 78.7% nt homology with YH-16 and an 89.2% aa homology with SH, which was a higher percentage than with other reference strains. For SMU-F4-2020, ORF1–3 shared 85.0%/95.5%, 83.5%/91.4%, and 86.3%/97.2% nt/aa homology levels with GX2019, GX01–13, and 12Q087–5, respectively, which were higher than the homology levels with other reference strains. For SMU-B5-2020, ORF1–3 shared their highest nt/aa homology levels with HRB-SS (81.9%/93.2%), FCV-21223 (83.2%/91.8%), and UTCVM-NH3 (91.3%/97.2%), respectively. Thus, recombination events among different viruses might have occurred in their genomes.

Furthermore, we constructed phylogenetic trees using the complete genomes to determine the genetic relationships of the three isolates. The complete genome of the SMU-B5-2020 strain belonged to cluster III of the GI group and shared a close genetic relationship with two USA strains, UTCVM-NH1 and 20879. The SMU-F4-2020 strain clustered in cluster IV of the GI group and shared close relationships with two Chinese strains, CH-JL4 (northeastern China) and GX2019 (southern China). In addition, the SMU-B22-2020 strain was the neighbor of YH-16 and CH-JL1 in the GII group (Figure 4). The results suggest that various FCV strains from different clusters are simultaneously circulating and spreading in southwestern China.

To identify possible recombinant events, we detected recombination using SimPlot and RDP4 software packages. The SMU-B5-2020 strain showed potential recombinant events between the FCV-21223 strain (cluster I of GI) and the HRB-SS strain (cluster III of GI) with *p*-values of <1 × 10^−6^ from the results of five detection methods. From the similarity plot, two recombination breakpoints within the SMU-B5-2020 genome were identified in ORF1 (nt 3104) and ORF2 (nt 5328) (Figure 5a). The breakpoints in SMU-B5-2020 separated its genome into three regions, with regions A (nt 1–3104) and C (nt 5328–7705) being closely related to the FCV-21223 strain, whereas region B (nt 3104–5328) was closely related to the HRB-SS strain (Figure 5b). The results indicated that the SMU-B5-2020 strain likely originated from recombination events between members of clusters I and III of the GI FCV strains.

### 3.5. Serum Neutralization Antibody Titers

Viral antigen was prepared by mixing inactivated FCVs with Freund’s complete adjuvant. A serum virus neutralization assay showed that the SMU-F4-2020-neutralizing antibody titer level in three mice reached as high as 1:775 ± 65 to the SMU-F4-2020 isolate, which was significantly higher than the titers to SMU-B22-2020, SMU-B5-2020, and the vaccine (*p* < 0.05). Similarly, the SMU-B22-2020-neutralizing antibody titer level reached 1:655 ± 55 to the SMU-B22-2020 strain, which was significantly higher than the titers to the other two isolates and the commercial vaccine (*p* < 0.05). The SMU-B5-2020-neutralizing antibody titer was 1:416 ± 7.5 to SMU-B5-2020, which was significantly higher than the titers to the other two isolates and the commercial vaccine (*p* < 0.05) (Table 2). The results indicated that the FCV-antibodies against each isolate did not provide better cross-reactivity between heterologous FCV isolates.

## 4. Discussion

In the general cat population, FCV infections are widespread and can cause respiratory disease, stomatitis, arthritis, and hemorrhagic-like fever [1]. At present, several types of vaccines are available for FCV. Although these commercial vaccines have been widely used in the field, FCV disease still remains a problem in cats. Vaccination against FCV has effectively reduced the clinical disease incidence but has not prevented highly virulent FCV strain (such as VS-FCV associated with VSD) outbreaks from occurring in vaccinated cats [6,9,25]. The frequent mutations of field FCV strains results in the poor protective effects of these vaccines. In this study, molecular epidemiological investigations and serum virus neutralization antibody assays of FCV in southwestern China were performed.

The prevalence of FCV is usually proportional to the number of cats in a household, with a higher infection rate usually occurring when large groups of cats are housed together. Therefore, pet cats kept in homes at low densities generally have a relatively low infection rate (approximately 10%) [26]. In contrast, cats living in animal shelters with large populations usually have a higher chance of being infected (approximately 25–40%) [3,27,28,29]. In China, the majority of FCV strains have been detected in domestic and stray cats [16,30]. To date, FCV strains from diseased cats were mainly reported in central and northeastern China [5,16,31,32], with limited molecular epidemiological information from southwestern China. In this study, 162 nasal swab samples collected from cats in southwestern China (Chengdu and Chongqing) were subjected to FCV detection. The total positivity rate was 23.46%, with cats in animal shelters having a significantly higher positive rate (35.71%) than those in households (10.26%), which is consistent with previous studies [3,7,27]. The result indicated that the infection rate is indeed associated with the number of cats.

VP1 is the most variable FCV structural protein and is composed of six regions (A–F). The variable region E contains major B-cell epitopes, which are targets of virus-neutralizing antibodies [33]. Thus, VP1 is often used as a target for clinical diagnoses and new vaccine development. Phylogenetic analysis using the *VP1* gene showed that FCV strains originating from southwestern China displayed abundant genetic diversity. All the detected FCV strains in this study clustered into two genogroups (GI and GII), and 8 out of 10 strains clustered into five clusters of GI, and the remaining 2 strains belonged to GII. Thus, the predominant viruses in southwest China were GI strains, which differs to a previous report in central China [5]. Full-length genome sequence analysis of three new FCV isolates (SMU-B22, SMU-F4, SMU-Y14, and SMU-B5) revealed that they shared 76.1–82.9% nucleotide identity levels with reference strains and had the HVs in the *VP1* gene. Phylogenetic analysis indicated that the three isolates were divided into different groups (two GI and one GII strains). Notably, the three FCV isolates in this study showed a distant genetic relationship with the vaccine strains FCV-2024 (AF479590) and F9 (M86379) based on whole-genome sequences, which suggested that the vaccines may not provide effective crossing protection against the epidemic FCV strains.

Recombination is pervasive among RNA viruses and is a major mechanism contributing to the emergence and evolution of new variant strains [34,35]. Recombination was observed in an FCV strain W112 from the UK [36] and a 14Q315 strain from the Republic of Korea [19]. In the present study, the SMU-B5-2020 strain showed evidence of recombination events between cluster I and III members of GI FCV strains, which is reported in China for the first time. Considering its parental strains are FCV 21223 from the USA and HRB-SS from Heilongjiang (northeastern China), we speculate that the SMU-B5-2020 strain may have undergone a complicated evolutionary process.

Mutation is another important mechanism of FCV evolution that leads to the genetic divergence of FCVs in the field [1]. However, it is unclear whether mutations of the FCVs will affect the immuno-protection provided by commercial vaccines. A recent study showed that antibodies raised to the FCV-F9 vaccine remain broadly cross-reactive to separate FCV field strains from six European countries as determined by an in vitro neutralization assay [21]. In this study, to evaluate the cross-reactivity between different FCV isolates in China, a serum virus neutralization assay was performed. The FCV-antibody provided greater antibody neutralization reactivity against homologous strains (titer 1:416.5 to 1:775) than heterologous strains (titer 1:19 to 1:106). In addition, the vaccine-neutralizing antibodies titer was 1:16 to FCV strains, which was lower than the neutralizing antibodies raised by FCV isolates. Additionally, due to the samples in this study being collected from vaccinated cats with obvious clinical symptoms, we speculate that the current commercial vaccine does not appear to provide enough cross-protection to the contemporary FCV epidemic strains in this study. Furthermore, the immuno-protection of current commercial vaccines needs to be further confirmed by animal experiments.

## 5. Conclusions

In summary, the present study describes the epidemiological characteristics of FCV strains isolated in southwestern China during 2017–2020, and the predominant strains were the GI type. One FCV isolate was found to originate from recombination events among different GI strains. Further serum virus neutralization assays suggested that FCV antisera had significantly different cross-reactivity levels to temporally separated FCV populations. Our study highlights the importance of continuously monitoring FCV strains and provides new insights into the application of vaccination strategies in China.

## Figures and Tables

**Figure 1 viruses-13-01812-f001:**
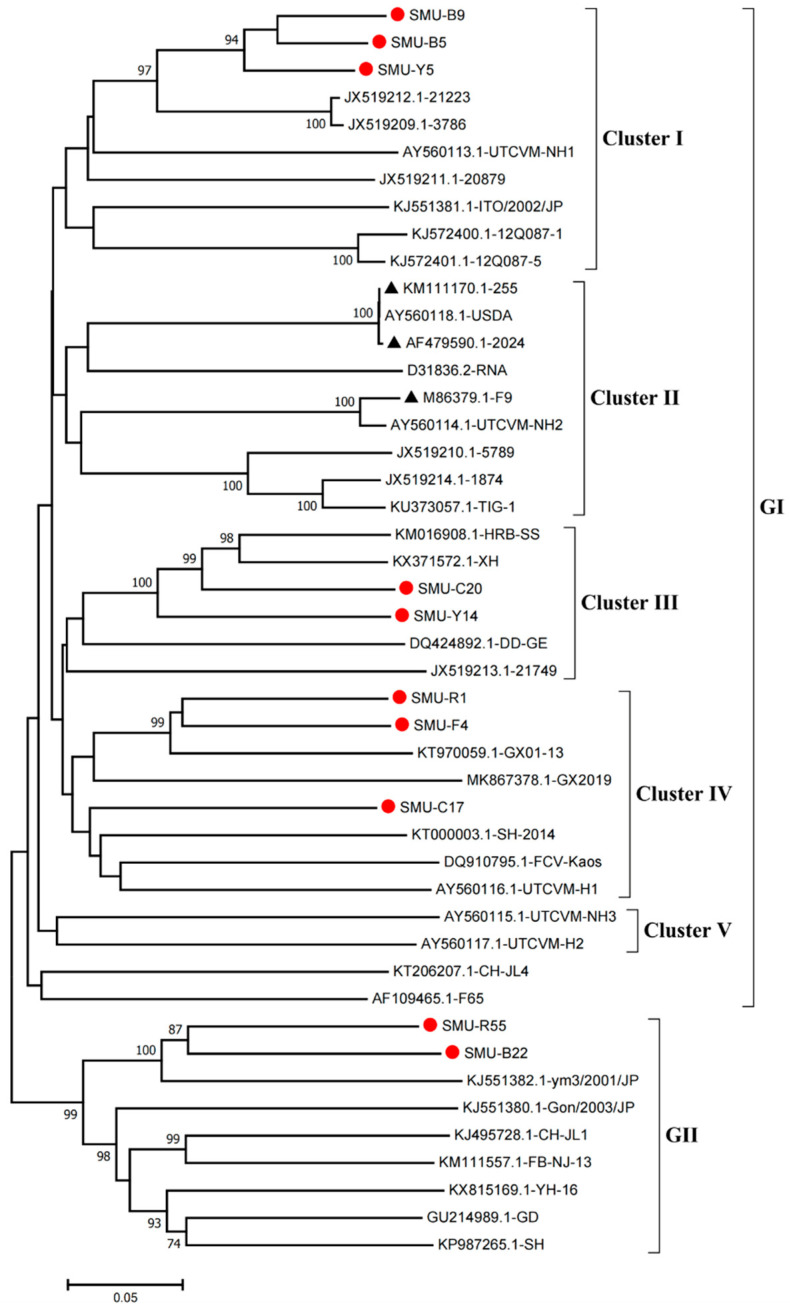
Phylogenetic tree of FCVs from southwestern China and 35 reference strains based on nucleotide sequences of VP1 HV. The phylogenetic tree was constructed with MEGA7.1 software (Tempe, AZ, USA) by the neighbor-joining method with 1000 bootstrap values. The viral assembly sequences characterized in this study are marked by “red circles” and the vaccine strains are labeled with “black triangles”. The scale bar indicates the number of nucleotide substitutions per site.

**Figure 2 viruses-13-01812-f002:**
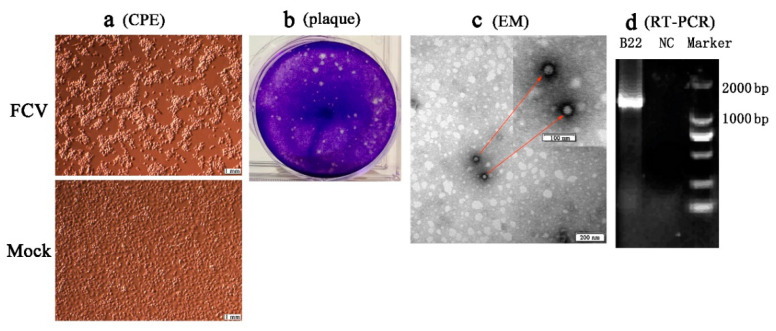
Isolation and identification of an FCV strain (SMU-B22-2020) in this study. (**a**) CRFK cells were inoculated with FCV SMU-B22-2020 and uninfected medium is shown at 12 h postinfection; (**b**) plaque assay for SMU-B22-2020 isolate with a dilution of 10^−10^ of virus stock using CRFK cells; (**c**) FCV particles with a diameter of approximately 30 nm were observed in cell supernatants by transmission electron microscopy (EM). Red arrows indicate the FCV particles; (**d**) the FCV isolate was further confirmed by RT-PCR with a 1300 bp target fragment. NC indicates the negative control.

**Figure 3 viruses-13-01812-f003:**
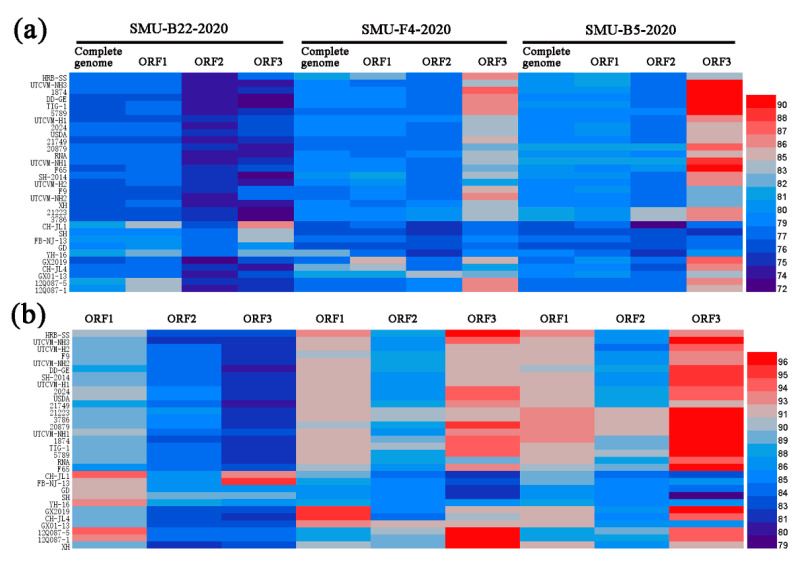
A heatmap was constructed based on the nucleotide sequence (**a**) and amino acid identities (**b**) of each genome among different FCV strains using HemI software. The colors represent sequence identity (blue for lower identities and red for higher identities) among different strains. The bar on the right represents the hierarchical similarity among the 15 meta-clusters (metric, Euclidean distance; linkage, average).

**Figure 4 viruses-13-01812-f004:**
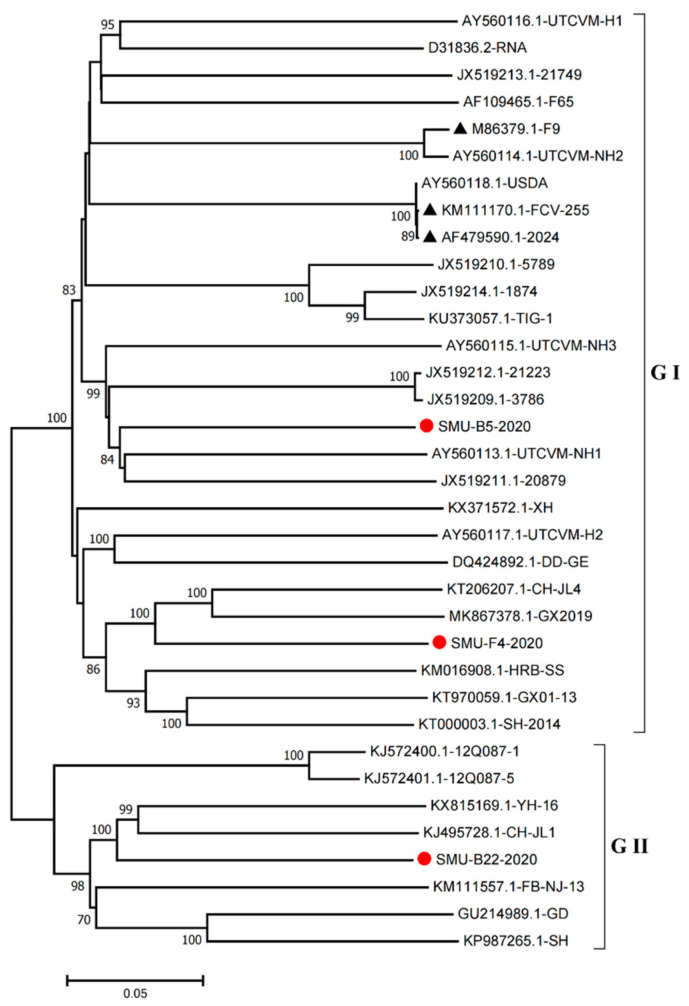
Phylogenetic tree based on full-length genomic sequence of three isolates (SMU-B22-2020, SMU-F4-2020, and SMU-B5-2020) and FCV reference strains available in GenBank. The three isolates in this study are labeled with “red circles” and the vaccine strains are labeled with “black triangles”. The scale bar indicates the number of nucleotide substitutions per site.

**Figure 5 viruses-13-01812-f005:**
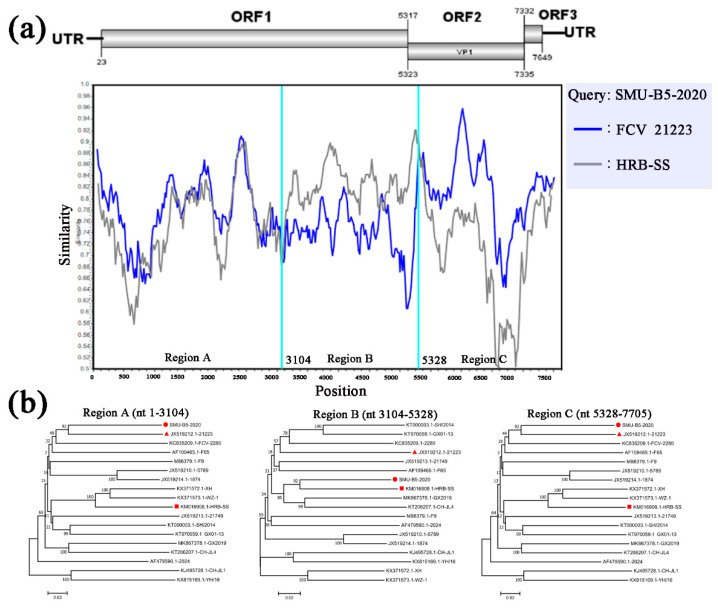
Full-length genome recombination analysis of SMU-B5-2020. (**a**) Genome scale similarity comparisons of SMU-B5-2020 (query) with FCV-21223 (blue) and HRB-SS (gray). The supposed recombination region is shown with red colors and the recombination breakpoints are marked at the bottom with nucleotide sites and viral genome structure referenced to FCV-21223; (**b**) phylogenetic trees based on every recombinant fragment within SMU-B5-2020 and 17 reference FCV strains are shown below the similarity plot. The strain SMU-B5-2020 is labeled with “red circles” and the putative recombinant parent strains are marked with “red triangles”.

**Table 1 viruses-13-01812-t001:** Information on the clinical specimens in this study.

No.	Collection Site	Year	No. of Samples	No. of Positives	Positive Rate (%)	Location	Clinical Sign
1	Chengdu	2017	5	5	100	Shelter	Conjunctivitis; nasal discharge
2	Chengdu	2017	5	0	0	Shelter	Nasal discharge
3	Chongqing	2018	8	4	50	Shelter	Conjunctivitis
4	Chengdu	2018	15	6	40	Shelter	Conjunctivitis
5	Chongqing	2018	11	3	27.3	Shelter	Conjunctivitis
6	Chengdu	2018	4	2	50	Shelter	Nasal discharge
7	Chongqing	2018	6	2	33.3	Shelter	Nasal discharge
8	Chengdu	2018	5	1	20	Shelter	Nasal discharge
9	Chengdu	2018	6	2	33.3	Shelter	Nasal discharge
10	Chengdu	2018	4	2	20	Shelter	Nasal discharge
11	Chongqing	2018	3	2	66.7	Shelter	Sneeze; nasal discharge
12	Chengdu	2019	3	1	33.3	Shelter	Nasal discharge
13	Chengdu	2019	5	0	0	Shelter	Sneeze; conjunctivitis
14	Chongqing	2019	4	0	0	Shelter	Sneeze; conjunctivitis
15	Chongqing	2020	20	2	10	Pet Hospital	Conjunctivitis; nasal discharge
16	Chengdu	2020	58	6	10.3	Pet Hospital	Conjunctivitis; nasal discharge
Total	-	-	162	38	23.46	-	-

**Table 2 viruses-13-01812-t002:** Determination of serum neutralizing antibody titers to FCV isolates in mice.

Neutralizing Antibody (NA)	FCV Isolates		
	SMU-F4-2020	SMU-B22-2020	SMU-B5-2020
SMU-F4-2020-NA	1:775 ± 65 ^a^	1:21.5 ± 2.5	1:19 ± 0
SMU-B22-2020-NA	1:106.6 ± 0.5	1:655 ± 55	1:63.5 ± 0.5
SMU-B5-2020-NA	1:95 ± 5	1:54.5 ± 9.5	1:416.5 ± 7.5
Vaccine-NA	1:16 ± 0	1:16 ± 0	1:16 ± 0

^a^ The neutralizing antibody titer is shown as mean ± standard deviation (SD).

## Data Availability

All sequences cloned in this study have been submitted to GenBank (GenBank accession no.: MW194992–MW195001 and MW194989–MW194991).

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
