# Peer review of "Molecular Characterization and Cross-Reactivity of Feline Calicivirus Circulating in Southwestern China"

_viruses, 2021, doi:10.3390/v13091812_

Round 1

Reviewer 1 Report

-

Author Response

Dear Reviewer,

Thank you for your comments concerning our manuscript, “Molecular characterization and cross-reactivity of feline calicivirus circulating in southwestern China (Manuscript ID: viruses-1367179).” These comments are valuable and very helpful for revising the manuscript and improving our subsequent studies.

Sincerely yours,

Bin Zhang (Professor, Ph.D)

Reviewer 2 Report

The paper need some revisions, all reported In the attached file

Author Response

Comments and Suggestions for Authors

This article investigated the prevalence and molecular characteristics of FCVs in southwestern China by analyzing 162 nasal swab samples collected from cats in animal shelters and pet hospitals, revealing a genetic diversity and complex cross-reactivity levels of FCVs and providing new insights related to vaccination strategies. Broad comments highlighting areas of strength and weakness. The study is original and well done and has an interesting potential from both epidemiological and vaccinal points of view, but in my opinion needs some minor corrections and a deepening of two topics: the FCV distribution worldwide (with which to compare the results) and the possibility of different results if MLV vaccines had been used (see specific comments).

Dear Reviewer,

Thank you for your comments concerning our manuscript, “Molecular characterization and cross-reactivity of feline calicivirus circulating in southwestern China (Manuscript ID: viruses-1367179).” These comments are valuable and very helpful for revising the manuscript and improving our subsequent studies.

The point-by-point responses to the comments are listed in the following revision notes.

Sincerely yours,

Bin Zhang (Professor, Ph.D)

Revision Notes

  1. Lines 58-59 (“Then, the isolated viruses were subjected to molecular characterization, and cross-reactivity experiments with FCV strains”): what does the second part of the sentence (and cross-reactivity experiments with FCV strains) mean? Perhaps that their cross-reactivity was evaluated? Specify better.

Response: Thanks for your questions. In our study, to evaluate the cross-reactivity between different FCV strains in China, four separated FCVs (SMU-B22, SMU-F4, SMU-Y14 and SMU-B5) that did not cluster phylogenetically were filtered out for further viral isolation, and then the FCV isolates was used to evaluate the cross-reactivity of neutralizing antibodies by the serum virus neutralization assay. To specify better, we revised the sentence as “In the present study, 162 FCV suspected clinical samples were collected from nasal swabs from cats in southwest China from 2017 to 2020, and then the molecular characterization, virus isolation and cross-reactivity of FCV strains were further investigated” in lines 58-61.

  1. Lines 70-72 (“All cats in this study were vaccinated with a triple-inactivated feline Rhinotracheitis-Calici-Panleukopenia-vaccine (Zoetis or Boehringer Ingelheim)”: since the vaccine brands are two (Zoetis and Boehringer Ingelheim), also the vaccines used to vaccinate cats should be two, while throughout the article you speak always only of one vaccine, as if the vaccine was always the same. How much vaccines were the cats vaccinated with?

Response: According to the instruction of the vaccine from Zoetis or Boehringer Ingelheim, the two companies use the same strains, including Feline rhinotracheitis virus (FRV-605 strain), Feline calicivirus (FCV-255 strain) and Feline panleukopenia virus (FPV-Cu-4 strain) as the triple-inactivated feline Rhinotracheitis-Calici-Panleukopenia-vaccine for production. The cats were intramuscularly injected with only one vaccine of Zoetis or Boehringer Ingelheim (1 mL/dose). In serum neutralization assay, the mice were intramuscularly injected with only vaccine from Zoetis in this study. Thus, we use only one vaccine from Zoetis in our study. To make it clear, we revised the sentence in lines 74-78 and 127-128.

  1. Line 120 (“mice were immunized with… the above triple-inactivated vaccine”: what does it means "above"? as just said, in the part related to cats (lines 70-72) the vaccines seem to be two (Zoetis and Boehringer Ingelheim, see previous comment), so if the vaccine brands are two, two are also the vaccines used: consequently, what is "the above vaccine"? Once this point has been clarified, also the summary must be modified (lines 20-21).

Response: In this study, we only used the one vaccine from Zoetis in serum neutralization assay. Thus, "the above vaccine" means the commercial triple-inactivated vaccine from Zoetis used in this study, which have been revised in revised manuscript.

  1. Lines 135-137 (“Thus, the cats in the animal shelters had a significantly higher positive rate than those in the pet hospitals”) and lines 275-277 (“In contrast, cats living in animal shelters with large populations usually have a higher chance of being infected (approximately 25%-40%) [3,27,28]”): there are many other papers that confirm this result and highlight the FCV diffusion in stray cats all over the world: it would be useful to add some of them. In the paper of Dall'Ara et al. [Dall’Ara P., Labriola C., Sala E., Spada E., Magistrelli S., Lauzi S. (2019): Prevalence of serum antibody titres against feline panleukopenia, herpesvirus and calicivirus infections in stray cats of Milan, Italy. Preventive Veterinary Medicine, 167, 32-38] there is a very useful table (Table 3) that resumes the results of the main studies conducted around the world.

Response: We agreed with your advice, and have added the reference [Dall’Ara P., Labriola C., Sala E., Spada E., Magistrelli S., Lauzi S. (2019): Prevalence of serum antibody titres against feline panleukopenia, herpesvirus and calicivirus infections in stray cats of Milan, Italy. Preventive Veterinary Medicine, 167, 32-38] in line 286, and the reference number has been modified throughout the article.

  1. Line 277 (“FCV stains”): the word “stains” must be correct in “strains”.

Response: Thanks for your carefulness. We have revised the word “stains” as “strains” in line 286.

  1. Line 293 (“which is differs from”): this sentence must be correct: “which is different” or “which differs”.

Response: We have revised the sentence“which is differs from” as “which differs” in lines 302-303.

  1. Line 296 (“76.1%-82.9% nt identity levels”): I don't think this sentence is correct… what is “nt”?

Response: The “nt” means “nucleotide”. To make it clear, we revised the abbreviation “nt” as “nucleotide” in line 305.

  1. Line 304 (“an FCV strain W112”): I don't think "an FCV" is correct (why n before F?)

Response: Thanks for your carefulness. We made a mistake and we have revised the word “an” as “a” in line 313.

  1. Lines 307-309 (“Because its parental strains are FCV 21223 from the USA and HRB-SS from Heilongjiang (northeastern China), suggesting that the SMU-B5-2020 strain may have undergone a complicated evolutionary process”): I think that "because" at the beginning of the phrase needs another continuation, not "suggesting".

Response: To make the sentence clear, we revised the sentence as “Considering its parental strains are FCV 21223 from the USA and HRB-SS from Heilongjiang (northeastern China), we speculate that the SMU-B5-2020 strain may have undergone a complicated evolutionary process” in lines 316-317.

  1. Line 322 (“we speculate that the current commercial vaccine does not provide enough immune-protection to manage the contemporary FCV epidemic strains in China”): first of all, what does it mean “to manage the contemporary FCV epidemic strains”? you can manage an epidemic, but not epidemic strains… And then, since you specified that cats were vaccinated with a killed vaccine of Zoetis or Boehringer, the used vaccines are two different, so the word "vaccine" should be plural (see comment lines 70-72). Furthermore, it must be specified " the current commercial vaccine used to vaccinate the tested cats", if not it seems that none of the vaccines on the market works.

Response: Firstly, to make a clear description of the sentence, we revised the sentence as “Additionally, due to the samples in this study were collected from vaccinated cats with obvious clinical symptoms, we speculate that the commercial vaccine does not appear to provide enough cross-protection to the current FCV epidemic strains in this study” in lines 332-333.

Secondly, “vaccine” has been revised as “vaccines” in line 73.

Lastly, we agreed with your suggestion, the sentence has been revised that “we speculate the current commercial vaccine does not appear to provide enough cross-protection to manage the contemporary FCV epidemic strains in this study”. Further, the immune-protection of current commercial vaccines needs to further confirm by animal experiments, which has added in revised manuscript in lines 334-335.

General considerations:

In my opinion, using MLV vaccines, the results could be different: I think that a discussion on this possibility must be added. Furthermore, it would have been interesting to have some more information on FCV positive cats (ie, age, clinical condition, vaccine protocols, etc.). Then, it would be better to clearly underlie that the strains isolated in vaccinated cats certainly cannot be related to used vaccines

Response: Thanks for your suggestions. Firstly, the cats in the study were immunized with inactivated vaccine, no MLV vaccines available in China. Secondly, the relevant information on age, weight, clinical signs and the vaccine protocols was added in lines 71-78. Lastly, to clarify the relationship between the FCV isolates in this study and vaccine strains, we added a sequence of the inactivated vaccine strain FCV-255 (GenBank no.: KM111170) to the phylogenetic analysis (see Fig.1 and 4). The result showed that no strains in this study were divided into cluster II of the GI group, which contains the representative vaccine strains FCV-255 (KM111170), FCV-2024 (AF479590) and F9 (M86379). Thus, the FCV strains identified in vaccinated cats certainly cannot be related to used vaccines. We have revised the sentence in lines 166-168.